# Geographic, Patient, and VA Medical Center Variation in the Receipt and Mode of Primary Care in a National Sample of Veterans with Diabetes during 2020

**DOI:** 10.3390/healthcare12060643

**Published:** 2024-03-13

**Authors:** Melanie Davis, Brian Neelon, John L. Pearce, Danira Medunjanin, Elizabeth Bast, Robert Neal Axon, Hermes Florez, Kelly J. Hunt

**Affiliations:** 1Charleston Health Equity and Rural Outreach Innovation Center (HEROIC), Ralph H Johnson VA Medical Center, Charleston, SC 29401, USAhuntke@musc.edu (K.J.H.); 2Department of Public Health Sciences, Medical University of South Carolina, Charleston, SC 29425, USA; 3Department of Ambulatory Medicine, Bruce W. Carter VA Medical Center, Miami, FL 33125, USA; elizabeth.bast@va.gov

**Keywords:** health services research, health geography, chronic disease management

## Abstract

While telemedicine infrastructure was in place within the Veterans Health Administration (VHA) healthcare system before the onset of the COVID-19 pandemic, geographically varying ordinances/closures disrupted vital care for chronic disease patients such as those with type 2 diabetes. We created a national cohort of 1,647,158 non-Hispanic White, non-Hispanic Black, and Hispanic veterans with diabetes including patients with at least one primary care visit and HbA1c lab result between 3.5% and 20% in the fiscal year (FY) 2018 or 2019. For each VAMC, the proportion of telehealth visits in FY 2019 was calculated. Two logistic Bayesian spatial models were employed for in-person primary care or telehealth primary care in the fourth quarter of the FY 2020, with spatial random effects incorporated at the VA medical center (MC) catchment area level. Finally, we computed and mapped the posterior probability of receipt of primary care for an “average” patient within each catchment area. Non-Hispanic Black veterans and Hispanic veterans were less likely to receive in-person primary care but more likely to receive tele-primary care than non-Hispanic white veterans during the study period. Veterans living in the most socially vulnerable areas were more likely to receive telehealth primary care in the fourth quarter of FY 2020 compared to the least socially vulnerable group but were less likely to receive in-person care. In summary, racial minorities and those in the most socially vulnerable areas were less likely to receive in-person primary care but more likely to receive telehealth primary care, potentially indicating a disparity in the impact of the pandemic across these groups.

## 1. Introduction

For chronic disease patients, such as those with Type 2 diabetes, primary care receipt is vital to ensure proper disease management. Continuity of primary care has been shown to increase the odds of adhering to diabetes monitoring guidelines by 36.0% for blood glucose monitoring and by 76.0% for the lipid profile test [1]. Furthermore, coordination of primary care among health workers and caregivers has been shown to improve patient outcomes such as hemoglobin A1c and blood pressure [2]. However, the COVID-19 pandemic has created unprecedented disruptions in care, changing if and how patients interact with their providers; thus, it is vital to investigate the impact of the pandemic on disruptions in care within this vulnerable patient population.

Chronic disease patients were more likely to miss primary care medical appointments during the pandemic when compared to pre-pandemic times [3]. Vulnerable patient populations were advised to isolate to reduce the risk of COVID-19. Hence, patient–provider care coordination required to provide medications, education and laboratory monitoring was reduced during the pandemic [4]. Clinicians remain concerned about continuity of care; as a result, downstream effects of care disruptions are still being studied [5]. Maintaining primary care and navigating medical care during the pandemic required telehealth infrastructure at the provider level and digital access and literacy at the patient level, underlining the need to address existing and emerging inequities in care [6].

It has been proposed that to address these inequities, social determinants of health must be included in research related to the COVID-19 pandemic [7]. Social and geospatial determinants of health include measures such as housing and household composition, socioeconomic status, and access to transportation, among others that describe a person’s natural and built environment. Social vulnerability, as defined by the Centers for Disease Control, refers to the “potential negative effects on communities caused by external stresses on human health” [8]. The emergence of COVID-19 qualifies as an external stressor and its impact on human health can only be fully understood by considering patient-, facility-, and geographic-level features of the population.

To investigate the impact of the pandemic on disruptions in care among a vulnerable chronic disease patient population, we examined the geographic, patient, and Veterans Affairs (VA) medical center (MC) characteristics associated with receiving primary care, and specifically mode of care (remote (telehealth) or in-person), among a cohort of veterans with Type 2 diabetes receiving care in the VA healthcare system. The VA is the nation’s largest provider of telehealth and had telemedicine infrastructure in place before the onset of the pandemic; however, community ordinances and closures required implementation and amplification of telehealth across the country. The first goal of our study was to present a comprehensive analysis of primary care during the pandemic in a vulnerable population: who was most likely to receive face-to-face primary care and who was receiving virtual care? The secondary goal of our study was to address inequities in COVID-19 research by incorporating social determinants of health measures; thus, we examined social vulnerability and regional effects in addition to demographics, disease burden, and VAMC characteristics.

## 2. Materials and Methods

Study Population: Veterans Health Administration (VHA) patient records in the Corporate Data Warehouse (CDW) were used to identify diabetic patients if they had two or more ICD-9/ICD-10 codes for diabetes (250.xx/E10.x, E11.x) across inpatient and outpatient records prior to the start of the study period. Requirements for cohort inclusions were having at least one valid HbA1c lab result (value: 3.5% to 20%); at least one primary care visit in fiscal year (FY) 2018 or 2019 (see primary care definition below); having a race–ethnicity of non-Hispanic White (NHW), non-Hispanic Black (NHB), or Hispanic (multiple (n = 8737), missing (n = 72,264) and other race-ethnicity groups (n = 34,039) were excluded); and having geocoded data on rurality and census tract in the VHA Planning Systems Support Group (PSSG) Geocoded Enrollee files [9]. This resulted in a final cohort of n = 1,647,158 diabetic veterans. The study was approved by our Institutional Review Board and local VA Research and Development committee.

Outcome Measure: We had two primary outcomes measures, both focused on receipt of primary care in the fourth quarter of 2020. We first derived “any type of primary care” in the fourth quarter of 2020 based on primary or secondary “Primary Care” stop code 323 excluding clinical pharmacy (secondary stop code 160), primary or secondary “Women’s Primary Care” stop code 322, primary “Group Primary Care” stop code 348, primary or secondary “Geriatric Primary Care” stop code 350, and secondary “Mental Health Primary Care” stop code 531. Our first outcome was receipt of in-person primary care in the fourth quarter of 2020, while our second outcome was receipt of virtual primary care in the fourth quarter of 2020 identified by remote care stop codes 338 and 683–686.

Primary Covariates: Covariates included (1) VA Medical Center (VAMC) catchment area, (2) quartile of CDC Social Vulnerability Index (SVI), and (3) quartile of VAMC telehealth implementation. Catchment areas, the geospatial units of analysis, were defined as the outer geographic boundary of counties in the vicinity of each of 125 VAMCs. Every US county was assigned to a catchment area based on which VAMC a plurality of patients from that county routinely received care from in FY 2003–2014. In our cohort, 12% of patients routinely received care outside their assigned catchment area based on county residence. As there were counties with split utilization, this mismatch was anticipated. Race and ethnicity were obtained from the outpatient and inpatient Medical SAS files and the PatSub race and ethnicity files [10]. We merged with our data the CDC’s publicly available SVI, a composite variable representing a census tract’s percentile ranking across four dimensions: socioeconomic status, household composition and disability, minority status and language, and housing type and transportation [11]. Patients were assigned an individual SVI value based on the SVI value of their residential census tract. Quartile of VAMC telehealth implementation was defined based on the proportion of telehealth visits completed in FY 2019 within a VAMC. Telehealth visits were defined based on the previously noted remote care stop codes.

Demographics and Comorbidity: Race–ethnicity, age, gender, marital status (married or not married), military service-connected disability dichotomized at 50% (cut-point veterans are exempt from VA co-payments), and comorbidity burden were available. Patients were classified as NHW, NHB, and Hispanic via patient report. Patients’ residential rurality was ascertained from the geocoded PSSG files (urban and rural/highly rural). The van Walraven algorithm for the weighted sum of Elixhauser comorbidities was applied based on ICD-10 codes, resulting in a continuous measure of comorbidity burden [12,13].

Medication Type and HbA1c level: Based on 2020 data, veterans were classified as taking insulin only, insulin and oral anti-hyperglycemic agents, or oral anti-hyperglycemic agents only. Additionally, as a measure of diabetes control prior to the pandemic onset, mean HbA1c was obtained for FY 2019.

Statistical Analysis: We estimated the effect of catchment area, quartile of CDC SVI and quartile of VAMC pre-pandemic telehealth implementation on receipt of primary care within the fourth quarter of 2020 using two logistic regression analyses to assess the two primary outcomes: (1) receipt of in-person primary care and (2) receipt of tele-primary care. Both models included demographics, comorbidity burden, medication use, HbA1c control level in 2019 (i.e., before the pandemic), quartile of CDC SVI, and quartile of VAMC pre-pandemic telehealth implementation. Each model included a spatial random effect for catchment area, allowing the model to capture spatial dependency among outcomes. The spatial random effect was assigned a conditional autoregressive (CAR) prior in a Bayesian setting to allow for features such as spatial smoothing, borrowing of information across catchments areas, and small area estimation [14].

All analyses were performed in R software 4.3.1 using a Gibbs sampling code we specifically developed to handle large, spatially correlated data [14,15]. For each model, we calculated the posterior probability of receipt of the particular type of primary care visit during fourth quarter of 2020 for an “average” patient within each catchment area (i.e., a patient with “average” covariate values). We subsequently mapped these probabilities to display the geographic distribution of primary care after controlling for the underlying population distributions as our goal was to isolate a spatial effect. Few catchment areas were impacted by small race strata (6% had between 6 and 30 Hispanic patients; 2% had between 10 and 30 NHB patients). The smoothing property of the spatial random effect ensured robust estimates even for sparse catchment areas. For both analyses, we assumed weakly informative priors for model parameters, allowing the data to play a dominant role in estimation. Additionally, we ran the Gibbs sampler for 5000 iterations following a “burn-in” to ensure convergence of the algorithm [16].

## 3. Results

The study population included 1,647,158 veterans with diabetes who received primary care at the VA in FY 2018 or 2019 and resided in 125 VAMC catchment areas, of which 72.50% were NHW, 21.61% were NHB, and 5.89% were Hispanic (Table 1). The mean age was 69.89 years, 95.14% were male, and 32.32% were not using oral medication or insulin to treat their diabetes. Figure 1 depicts national SVI values at the census tract level.

### Spatial Variation across Catchment Areas

In model 1, we adjusted for age, gender, race, rural–urban residence, service-connected disability, marital status, van Walraven comorbidity score, diabetes medication use and type, diabetes control prior to the pandemic, quartile of the CDC SVI, pre-pandemic VAMC telehealth adoption, and spatial random effects for catchment areas. Figure 2a provides a probability map of having an in-person primary care visit by catchment area for the study population based on parameter estimates, with shading reflecting adjusted quintile of probability estimates. Results from our adjusted logistic regression analysis revealed that the prevalence of having an in-person primary care visit during the fourth quarter of 2020 varied substantially by VAMC catchment area, with values ranging from 16% to 51%. The legend reflects the catchment areas with the highest and lowest probability of any primary care within each quintile. The spatial random effect variance was 0.59 (Table 2), indicating the presence of substantial unexplained variation.

Model 2 included the same covariates as model 1, with the outcome for model 2 being tele-primary care. Figure 2b provides probability maps of having a tele-primary care visit by catchment area for the study population based on parameter estimates, with shading reflecting adjusted quintile of probability estimates. Results from our adjusted logistic regression analyses revealed that the probability of having a tele-primary care visit during the fourth quarter of 2020 ranged from 14% to 52% depending on VAMC catchment area. There was substantial unexplained variation with a spatial random effect variance of 0.70 for the tele-primary care model (Table 2).

## 4. Discussion

We sought to examine geographic, patient-level, and VA-medical center variation in the receipt of in-person versus tele-primary care during the COVID-19 pandemic. Our cohort of veterans receiving care in the VA prior to the pandemic presented a unique opportunity to study a vulnerable population within a healthcare system that had the necessary existing infrastructure for delivering telemedicine during an unprecedented global pandemic.

Previously, we modeled the receipt of any primary care in the fourth quarter of fiscal year 2020 and observed that after adjusting for covariates, there remained substantial spatial variation in receipt of any form of primary care [17]. Hence, other geographically varying factors were associated with receipt of primary care. We also observed that as social vulnerability increased, the probability of receiving any primary care also increased. This result indicates that patients living in areas with socioeconomic, housing and transportation risk factors that put them at risk during public health emergencies (CDC SVI) were more likely to receive primary care than their peers living in areas without these risk factors.

In the current analysis, we analyzed the probability of receiving an in-person primary care visit and a virtual (i.e., tele) primary care visit, respectively. In both analyses, we observed that unexplained geographic variability remained after adjusting for social vulnerability, VAMC telehealth implementation in 2019, and patient-level factors. We observed that as social vulnerability increased, the probability of receiving in-person primary care decreased, while the probability of receiving tele-primary care increased. This result indicates that patients living in areas with socioeconomic, housing and transportation risk factors that put them at risk during public health emergencies (CDC SVI) were less likely to receive in-person primary care but more likely to receive tele-primary care than their peers living in areas without these risk factors. We also observed that NHB and Hispanic veterans were less likely than NHW veterans to receive in-person primary care but more likely to receive tele-primary care in the fourth quarter of fiscal year 2020. Rural veterans were also more likely than urban veterans to receive in-person primary care but less likely to receive tele-primary care. This is consistent with recent studies that also looked at the association between race and rurality and the receipt of telecare [18]. Additionally, we found that lower pre-pandemic VAMC telehealth implementation was associated with a lower overall probability of receiving tele-primary care during the fourth quarter of 2020. This difference is likely attributed to slower telehealth deployment in these VAMCs with less telehealth experience prior to the pandemic, and thus the significantly lower probability of telehealth visits noted in this cohort; it is worth highlighting that these VAMC were also less likely to provide in-person care. Our study is limited to VA services only and does not currently look at care received outside of the VA.

Additional limitations of our study include reduced generalizability, as the veteran population is not representative of the broader US population; this is because there is limited female representation; potential bias due to a broad definition of Type 2 diabetic veterans based on diagnosis codes alone; which allows for inclusion of veterans whose conditions are managed through lifestyle modification or who otherwise have not yet begun a medication regimen and who would be excluded based on more stringent and conservative criteria; and a narrow follow-up period for our outcome measures, suggesting that as additional and more recent data are made available. On this basis, our analyses could be expanded.

## 5. Conclusions

In summary, the results of this study provide a profile of patients with diabetes who did not receive face-to-face primary care during the pandemic: they are urban-dwelling, socially vulnerable, and racially diverse. However, the reasons for this—illness, avoidance of in-person contact, etc.—cannot be assessed in this study. As there is evidence that patients prefer in-person visits, future studies examining persistent post-pandemic barriers to access and the long-term impacts of non-preferential modes of care will be vital [19]. Lastly, the observed differences in modes of primary care during the pandemic may translate to changes in patient wellness behaviors such as treatment non-adherence, missingness of labs and other metrics, and loss of follow-up and specialty care in addition to health outcomes such as hospitalization and mortality. The next phases of our work aim to answer questions about post-disruption patterns of care, health outcomes, and the persistence of disparities and ultimately seek to lend guidance to the medical community and VA decision makers.

## Figures and Tables

**Figure 1 healthcare-12-00643-f001:**
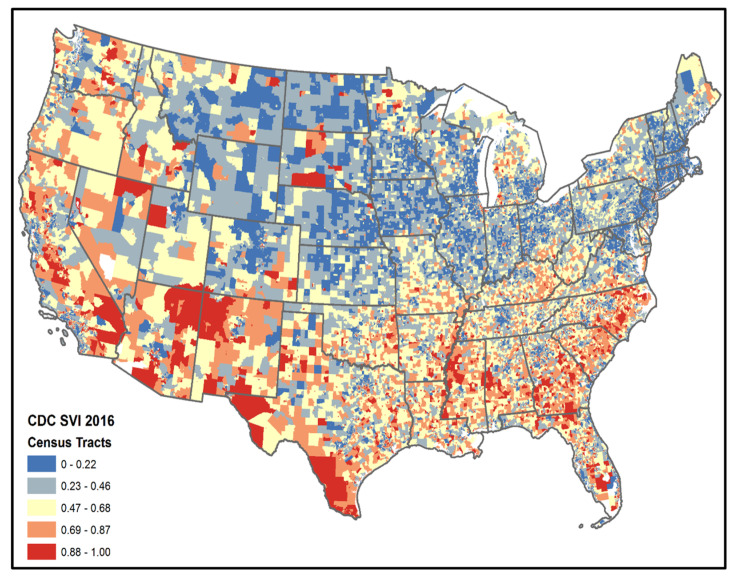
CDC Social Vulnerability Index based on 2016 census tracts.

**Figure 2 healthcare-12-00643-f002:**
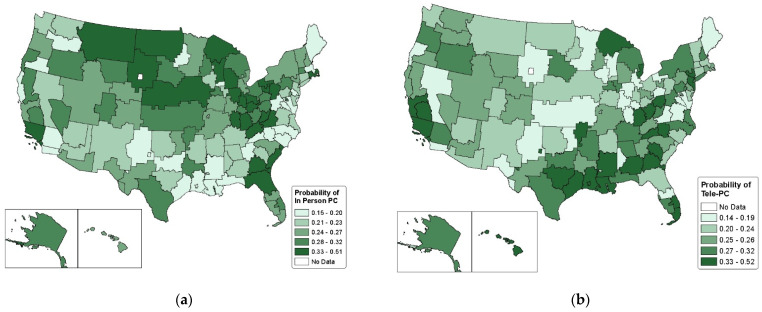
Adjusted predicted probability of receipt of in-person primary care (**a**) and tele-primary care (**b**) by catchment area. Predictions are based on Models 1 and 2 in Table 2 and correspond to individuals with average covariate values for all variables included in the model (age, gender, race, rural–urban residence, marital status, service-connected disability, van Walraven comorbidity score, diabetes medication use, mean HbA1c levels in 2019, CDC SVI, and pre-pandemic VAMC telehealth implementation.

**Table 1 healthcare-12-00643-t001:** Demographic features of the cohort.

		Overall	Social Vulnerability Q1	Social Vulnerability Q2	Social Vulnerability Q3	Social Vulnerability Q4
n		1,647,158	316,086	452,403	484,048	392,783
Age (Mean (SD))		69.89 (11.32)	71.10 (11.54)	70.27 (11.44)	69.69 (11.21)	68.73 (10.99)
Male		95.14	95.90	95.37	95.00	94.41
Race						
	non-Hispanic White	72.50	83.53	80.50	73.76	52.90
	non-Hispanic Black	21.61	12.58	14.90	20.74	37.79
	Hispanic	5.89	3.89	4.60	5.50	9.31
Rural		37.43	27.53	39.80	45.54	32.62
Married		59.98	67.66	63.85	59.51	49.94
Service-Connected Disability ≥ 50		37.60	38.42	38.43	38.05	35.45
van Walraven Comorbidity Score(Mean (SD))		4.16 (7.69)	4.06 (7.39)	4.08 (7.51)	4.17 (7.69)	4.32 (8.09)
Medications						
	No Medications					
	Orals	34.21	33.41	34.20	34.63	34.36
	Insulin	8.80	8.83	8.65	8.77	8.99
	Orals and Insulin	24.67	22.45	23.82	25.18	26.77
HbA1c in 2019 (Mean (SD))		7.20 (1.42)	7.13 (11.54)	7.17 (1.37)	7.21 (1.43)	7.25 (1.54)
VAMC Telehealth Visits FY19						
	Quartile 1 (highest proportion > 28%)	24.49	23.06	22.32	24.50	28.17
	Quartile 2 (23–28% telehealth)	21.53	22.48	20.49	20.52	23.36
	Quartile 3 (16–22% telehealth)	26.09	27.92	28.50	25.65	22.27
	Quartile 4 (lowest proportion < 16%)	27.89	26.53	28.70	29.34	26.19

**Table 2 healthcare-12-00643-t002:** Posterior odds ratios (95% credible intervals) for receipt of primary care in the fourth quarter of FY 2020 among diabetic veterans using VA services in the FY 2018–2019 period (n = 1,647,158).

		In-Person Primary Care	Tele-Primary Care
Age (1 year)		0.99 (0.99, 0.99)	0.99 (0.99, 0.99)
Male (female)		0.78 (0.77, 0.79)	0.77 (0.76, 0.79)
Race			
	non-Hispanic White	1.00	1.00
	non-Hispanic Black	0.81 (0.81, 0.82)	1.21 (1.20, 1.22)
	Hispanic	0.89 (0.88, 0.90)	1.16 (1.14, 1.18)
Rural (Urban)		1.10 (1.09, 1.11)	0.91 (0.90, 0.92)
Married (not married)		1.00 (0.98, 1.00)	0.94 (0.93, 0.95)
Service-Connected Disability ≥ 50		1.20 (1.22, 1.24)	1.24 (1.23, 1.26)
van Walraven Comorbidity Score		1.01 (1.01, 1.01)	1.01 (1.01, 1.01)
Medications			
	No Medications	1.00	1.00
	Orals	1.43 (1.61, 1.63)	1.39 (1.38, 1.41)
	Insulin	1.26 (1.29, 1.32)	1.40 (1.38, 1.41)
	Orals and Insulin	1.77 (1.75, 1.78)	1.82 (1.80, 1.84)
Mean HbA1c in 2019		0.94 (0.93, 0.94)	0.95 (0.95, 0.95)
CDC Social Vulnerability Index			
	Quartile 1 (Least vulnerable)	1.00	1.00
	Quartile 2	1.01 (1.00, 1.02)	1.07 (1.06, 1.08)
	Quartile 3	0.98 (0.97, 0.99)	1.18 (1.17, 1.20)
	Quartile 4 (Most vulnerable)	0.96 (0.94, 0.97)	1.27 (1.26, 1.29)
VAMC Telehealth Visits FY19			
	Quartile 1 (highest proportion > 28%)	1.00	1.00
	Quartile 2 (23–28% telehealth)	0.91 (0.90, 0.92)	0.71 (0.71, 0.72)
	Quartile 3 (16–22% telehealth)	0.99 (0.99, 1.00)	0.56 (0.55, 0.57)
	Quartile 4 (lowest proportion < 16%)	0.93 (0.92, 0.94)	0.39 (0.39, 0.39)
Spatial Variation		0.59	0.70

## Data Availability

Data are unavailable due to privacy or ethical restrictions.

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
