# Peer review of "Geographic, Patient, and VA Medical Center Variation in the Receipt and Mode of Primary Care in a National Sample of Veterans with Diabetes during 2020"

_healthcare, 2024, doi:10.3390/healthcare12060643_

Round 1

Reviewer 1 Report

Comments and Suggestions for Authors

This is a large retrospective study that aimed to analyze the impact of social vulnerability and geographic factors on telemedicine use in the Veteran Affairs healthcare system. The interest in this topic is high because identifying barriers to implementing telemedicine and addressing them would increase the quality of the medical system.

Introduction and abstract

The abbreviations should be explained at their first use: FY, VA, NHB, NHW.

The scope of the study is formulated as a conclusion at the end of the Introduction. I suggest reformulating R63-66.

Materials and methods

R97: Figure 1 belongs to the Results section. I suggest deleting this sentence.

R110: How did you handle patients that had treatment modification, eg, from oral antidiabetics to insulin?

Please specify why you chose 3,5% as the lower limit. In previous studies, the lower limit was 4,0% which corresponds to an average of 68 mg/dl. (Nathan DM, Kuenen J, Borg R, et al. Translating the A1C assay into estimated average glucose values [published correction appears in Diabetes Care. 2009 Jan;32(1):207]. Diabetes Care. 2008;31(8):1473-1478. doi:10.2337/dc08-0545)

Discussion

R195: I suggest moving the limitation paragraph at the end of the discussion section

You specified in results that 35% of patients did not have medication and you included patients with HbA1c 3,5%. Could some patients be misclassified as having diabetes? I suggest adding this to the limitations.

I did not identify a specific conclusion that is based on the results.  The authors should present the profile of the vulnerable patient based on their results that can be generalized only to men (95% of the population).

Author Response

The abbreviations should be explained at their first use: FY, VA, NHB, NHW.

This has now been addressed. 

The scope of the study is formulated as a conclusion at the end of the Introduction. I suggest reformulating R63-66.

We agree and have reformulated this section.

Materials and methods

R97: Figure 1 belongs to the Results section. I suggest deleting this sentence.

We thank the reviewer for this suggestion and have deleted the sentence accordingly.

R110: How did you handle patients that had treatment modification, eg, from oral antidiabetics to insulin?

Patients receiving both oral medications and insulin in the study period were classified as having had "both" medications.

Please specify why you chose 3,5% as the lower limit. In previous studies, the lower limit was 4,0% which corresponds to an average of 68 mg/dl. (Nathan DM, Kuenen J, Borg R, et al. Translating the A1C assay into estimated average glucose values [published correction appears in Diabetes Care. 2009 Jan;32(1):207]. Diabetes Care. 2008;31(8):1473-1478. doi:10.2337/dc08-0545)

We have used 3.5% as the lower cutoff in all work associated with this cohort so as to exclude unlikely/implausible values for A1c. This threshold is used as inclusion criteria in this study and the rationale behind using a different threshold may vary depending on the use of the value (covariate, outcome measure, etc.) for individual studies and their aims.

Discussion

R195: I suggest moving the limitation paragraph at the end of the discussion section

We have moved the limitations to the second to last paragraph in the Discussion and have incorporated suggestions across reviewers to highlight the unique limitations of our study.

You specified in results that 35% of patients did not have medication and you included patients with HbA1c 3,5%. Could some patients be misclassified as having diabetes? I suggest adding this to the limitations.

This has been incorporated into the limitations paragraph referenced above.

I did not identify a specific conclusion that is based on the results.  The authors should present the profile of the vulnerable patient based on their results that can be generalized only to men (95% of the population).

The final paragraph of the manuscript has been edited to present a profile consistent with our findings. We emphasized the generalizability issue in the limitations paragraph (second to last). 

Reviewer 2 Report

Comments and Suggestions for Authors

The paper emphasizes how important it is to comprehend how the epidemic has affected the provision of healthcare, particularly for vulnerable groups. Though differences in telehealth consumption occurred, it implies that telemedicine played a significant part in maintaining treatment during the epidemic, highlighting the need for equitable healthcare access and policy considerations.

The research highlights the importance of telehealth and socioeconomic determinants of health while offering insightful information about how healthcare delivery is changing amid a public health emergency. To investigate the long-term effects of these discrepancies on patient outcomes and healthcare policy, more investigation is required.

The authours are requested to address the following points to enhance the paper presentation:

Introduction:

Provide a brief context or background on the significance of investigating the impact of the COVID-19 pandemic on healthcare delivery, particularly for chronic disease patients.

Identify the research aims and questions you intend to address in the study.

Data and results:

Explain the cohort's selection criteria more explicitly. You described the inclusion criteria briefly, but including a clear and straightforward table or flowchart detailing the cohort selection process would be beneficial.

Explain how the CDC Social Vulnerability Index (SVI) was generated and how it relates to the study.

Mention the limits of employing VA data and the potential consequences for generalizability.

Discuss the implications of your findings. What impact do discrepancies in in-person vs. virtual care have on the health outcomes of diabetic veterans?

Discuss the limits of your study in further detail, including any potential sources of bias or confounding variables.

Consider discussing the practical consequences of your findings with healthcare policymakers and providers.

Conclusion:

Summarize the key results and their potential implications in the context of the COVID-19 pandemic.

Highlight any recommendations for future study or changes to healthcare policy based on your results.

Comments on the Quality of English Language

The paper emphasizes how important it is to comprehend how the epidemic has affected the provision of healthcare, particularly for vulnerable groups. Though differences in telehealth consumption occurred, it implies that telemedicine played a significant part in maintaining treatment during the epidemic, highlighting the need for equitable healthcare access and policy considerations.

The research highlights the importance of telehealth and socioeconomic determinants of health while offering insightful information about how healthcare delivery is changing amid a public health emergency. To investigate the long-term effects of these discrepancies on patient outcomes and healthcare policy, more investigation is required.

The authours are requested to address the following points to enhance the paper presentation:

Introduction:

Provide a brief context or background on the significance of investigating the impact of the COVID-19 pandemic on healthcare delivery, particularly for chronic disease patients.

Identify the research aims and questions you intend to address in the study.

Data and results:

Explain the cohort's selection criteria more explicitly. You described the inclusion criteria briefly, but including a clear and straightforward table or flowchart detailing the cohort selection process would be beneficial.

Explain how the CDC Social Vulnerability Index (SVI) was generated and how it relates to the study.

Mention the limits of employing VA data and the potential consequences for generalizability.

Discuss the implications of your findings. What impact do discrepancies in in-person vs. virtual care have on the health outcomes of diabetic veterans?

Discuss the limits of your study in further detail, including any potential sources of bias or confounding variables.

Consider discussing the practical consequences of your findings with healthcare policymakers and providers.

Conclusion:

Summarize the key results and their potential implications in the context of the COVID-19 pandemic.

Highlight any recommendations for future study or changes to healthcare policy based on your results.

Author Response

Introduction:

Provide a brief context or background on the significance of investigating the impact of the COVID-19 pandemic on healthcare delivery, particularly for chronic disease patients.

In order to tie together the background provided in the first paragraph and the subsequent introductory paragraphs and the goal of our study, we amended lines 39-41 and 60-61.

Identify the research aims and questions you intend to address in the study.

We appreciate the reviewers remarks and have rephrased lines 67-72 to more clearly state the goals of our study.

Data and results:

Explain the cohort's selection criteria more explicitly. You described the inclusion criteria briefly, but including a clear and straightforward table or flowchart detailing the cohort selection process would be beneficial.

We have edited the Study Population section to more explicitly describe the cohort selection.

Explain how the CDC Social Vulnerability Index (SVI) was generated and how it relates to the study.

We have edited the Primary Covariates section to better describe the SVI and its inclusion in the study.

Mention the limits of employing VA data and the potential consequences for generalizability.

We have now addressed these (95% male, limited to only VA care...) in the limitations in the Discussion section.

Discuss the implications of your findings. What impact do discrepancies in in-person vs. virtual care have on the health outcomes of diabetic veterans?

We appreciate this insightful comment and have expanded our final paragraph to discuss the implications and future directions more thoroughly.

Discuss the limits of your study in further detail, including any potential sources of bias or confounding variables.

We now discuss the limitation of our broad definition of the diabetic cohort, i.e. not requiring patients to also be on prescription medication in the second to last paragraph of the Discussion. We are thankful to the reviewer for this suggestion.

Consider discussing the practical consequences of your findings with healthcare policymakers and providers.

We now touch on this in the final summary paragraph and will (hopefully) be able to suggest more practical consequences and recommendations in future works.

Conclusion:

Summarize the key results and their potential implications in the context of the COVID-19 pandemic.

We now summarize our results more clearly by highlighting the patient profile most at risk for not having received in-person primary care (final paragraph, first sentence).

Highlight any recommendations for future study or changes to healthcare policy based on your results.

We now explain the future directions and how they could inform healthcare policy (final paragraph, final two sentences).

Reviewer 3 Report

Comments and Suggestions for Authors

The study, "Geographic, Patient, and VA Medical Center Variation in Receipt and Mode of Primary Care in a National Sample of Veterans with Diabetes during 2020," conducted by Davis et al., aims to elucidate the diverse modes of primary care access for veterans with type 2 diabetes, emphasizing distinctions between urban and rural areas as well as in different race.

Several points need to be addressed before considering for the publication.

1.       The authors exclusively concentrate on type 2 diabetes without addressing type 1 diabetes among veteran patients. Notably, type 1 diabetes is often considered more fatal. What factors contribute to this distinction?

2.       What was the reason for Non-Hispanic Black veterans and Hispanic veterans were less likely to receive in-person primary care than non-Hispanic white veterans during the study period?

3.       In the Materials and Methods section, the authors indicated that patients were categorized as Non-Hispanic White (NHW), Non-Hispanic Black (NHB), and Hispanic based on patient self-report. The analysis excluded multiple (n=8,737), missing (n=72,264), and other race-ethnicity groups (n=34,039). Could you provide clarification on why the other race-ethnicity groups were excluded from the analysis?

4.       In the Results section, the authors mentioned that 32.32% of patients were not utilizing oral medication or insulin for diabetes treatment. The inquiry arises: how do these patients effectively manage their diabetes without these conventional treatments?

5.       The authors specifically emphasized data exclusively from male veterans and did not include information from female veterans. Could you provide insights into the rationale behind this gender-specific focus?

Author Response

1. The authors exclusively concentrate on type 2 diabetes without addressing type 1 diabetes among veteran patients. Notably, type 1 diabetes is often considered more fatal. What factors contribute to this distinction?

We acknowledge the importance of studying Type I diabetes; the majority of veterans with diabetes are living with Type 2 diabetes and thus allow for a robust cohort size. Type 1 and Type 2 patients are distinct in the risk factors, onset, and management of their disease and thus should not be combined in a study with goals such as ours. We agree that the impact of the pandemic should be carefully studied in the unique Type I population.

2. What was the reason for Non-Hispanic Black veterans and Hispanic veterans were less likely to receive in-person primary care than non-Hispanic white veterans during the study period?

Our study (unfortunately) cannot answer this and we do state this in the final paragraph of the manuscript; however, as our analyses are adjusted for many patient- and several environmental-level factors, the evidence suggests that the disparity persists even after controlling for these factors (e.g. urban/rural residence or disease severity). The reasons may include a complex mixture of systemic features and personal barriers that are difficult to ascertain from patient health records and aggregate community data. We are currently expanding from the current study and working to understand the impact of the observed disparity on post-disruption patterns of care and health outcomes such as hospitalizations and mortality and hope that the conclusions from those studies can be relayed to policy makers and more directly impact at-risk NHB and Hispanic veterans.

  1. In the Materials and Methods section, the authors indicated that patients were categorized as Non-Hispanic White (NHW), Non-Hispanic Black (NHB), and Hispanic based on patient self-report. The analysis excluded multiple (n=8,737), missing (n=72,264), and other race-ethnicity groups (n=34,039). Could you provide clarification on why the other race-ethnicity groups were excluded from the analysis?

NHW, NHB, and Hispanic veterans comprise the vast majority of racial groups within the VA. There have been studies that specifically aim to best classify and describe the "multiple" indicated race groups and other studies that have proposed methodology to impute the missing race data. This was beyond the scope of our current study. The "other" group is a mixture of diverse races - Pacific Islander, Asian, Alaska Native, etc. and we find, especially when conducting studies with a geographic component, that the subgroups are not geographically widespread and do not allow for robust or precise estimation of effects. 

  1. In the Results section, the authors mentioned that 32.32% of patients were not utilizing oral medication or insulin for diabetes treatment. The inquiry arises: how do these patients effectively manage their diabetes without these conventional treatments?

We now highlight (in the second to last paragraph of the Discussion) the inclusion of these veterans who have 2 ICD-10 codes but are not on medications as a potential limitation of our study and suggest that they may have been "prescribed" lifestyle modifications or have not yet begun a VA medication regimen.

5. The authors specifically emphasized data exclusively from male veterans and did not include information from female veterans. Could you provide insights into the rationale behind this gender-specific focus?

We now highlight the limitation of generalizability given the high proportion of male veterans (second to last paragraph of the Discussion). This is a common and nearly unavoidable consequence of utilizing VA data: there are many more male veterans than female veterans especially among older cohorts. It was certainly not our focus . Our study benefited from an overall large cohort size (>1.6 million), allowing us to estimate the effect of sex with a reasonable interval; however, we acknowledge that the broader US patient population is more balanced than the 95:5 male to female ratio in our data.

Round 2

Reviewer 2 Report

Comments and Suggestions for Authors

The authors address all of the raised issues.